# Fast and Sensitive Analysis of Short- and Long-Chain Perfluoroalkyl Substances in Foods of Animal Origin

**DOI:** 10.3390/molecules27227899

**Published:** 2022-11-15

**Authors:** Federica Gallocchio, Alessandra Moressa, Gloria Zonta, Roberto Angeletti, Francesca Lega

**Affiliations:** Istituto Zooprofilattico Sperimentale delle Venezie, Chimica, 35020 Legnaro, Padova, Italy

**Keywords:** PFAS, GenX, C6O4, food of animal origin, QuEChERS, LC-MSMS

## Abstract

The availability of sensitive analytical methods to detect per- and polyfluoroalkyl substances (PFASs) in food of animal origin is fundamental for monitoring programs to collect data useful for improving risk assessment strategies. The present study aimed to develop and validate a fast and sensitive method for determining short and long-chain PFASs in meat (bovine, fish, and swine muscle), bovine liver, hen eggs, and cow’s milk to be easily applicable in routine analysis of food. A QuEChERS extraction and clean-up method in combination with liquid chromatography coupled to mass spectrometry (LC-MSMS) were used. The method resulted in good linearity (Pearson’s R > 0.99), low limits of detection (7.78–16.35 ng/kg, 8.26–34.01 ng/kg, 6.70–33.65 ng/kg, and 5.92–19.07 ng/kg for milk, liver, egg, and muscle, respectively), and appropriate limits of quantification (50 ng/kg for all compounds except for GenX and C6O4, where the limits of quantification were 100 ng/kg). Trueness and precision for all the tested levels met the acceptability criteria of 80–120% and ≤20%, respectively, regardless of the analyzed matrix. As to measurement uncertainty, it was <50% for all compound/matrix combinations. These results demonstrate the selectivity and sensitivity of the method for simultaneous trace detection and quantification of 14 PFASs in foods of animal origin, verified through the analysis of 63 food samples.

## 1. Introduction

Per- and polyfluoroalkyl substances (PFASs) are a class of contaminants characterized by a chemical structure based on a carbon chain that is fully or partially fluorinated [1,2]. The carbon–fluorine bonds are responsible for the peculiar and unique characteristics of these compounds that show high thermal, chemical, and biochemical stability as well as both hydrophobic and oleophobic properties [1]. Since the 1940s, the versatility of PFASs has been exploited in different fields including textiles, carpet and leather treatment, surfactants, firefighting foams, metal plating, and paper grease-proofing treatments [1].

The concern over possible risks of PFASs to the environment and human health remained unknown until the beginning of the new millennium, when two groups of PFASs, long-chain (LC) perfluoroalkyl carboxylic acids (PFCAs, C ≥ 7) and long-chain perfluoroalkane sulfonic acids (PFSAs, C ≥ 8) were shown to be ubiquitously present in biota and humans [3] and to have hazardous properties. In particular, perfluoro ottanoic acid (PFOA) and perfluoro octano sulfonate (PFOS) are the two LC PFAS (PFCA and PFSA, respectively) most often studied and reported in the scientific literature because of their massive production in the past as well as their widespread presence in the environment [4].

Consequently, global measures to ban or reduce the emission of LC PFAS have been implemented [5]. PFOS and PFOA are listed as persistent organic pollutants (POPs) under the Stockholm Convention. In the EU, proposals to place the PFCAs on the candidate list of Substances of Very High Concern (SVHC), are ongoing, but to date, only PFOA, its salts, and related substances have been included [5,6]. These restrictions recently led to the replacement of LC PFASs with short-chain (SC) PFASs, whose toxicological properties are still under study but have raised significant concerns [5].

Up-to-date studies have shown that SC PFASs might be as persistent as LC PFASs, with different, but no less alarming, properties of concern; however, little evidence of their distribution in the environment is available yet [7,8]. This is, for example, the case with two compounds, GenX and C6O4, recently detected in rivers and drinking waters [6,9,10,11,12]. GenX was introduced almost 10 years ago as a replacement for PFOA, and the occurrence of GenX in surface water at sampling sites located downstream of industrial areas has been recently documented [9,10]. Only two studies have documented the concentration of GenX in drinking water [6,10], but toxicological data are still incomplete and scarce [9]. Regarding C6O4, there are no published studies, and one of the main drawbacks in the detection of this compound is the unavailability of analytical standards. It has been detected in the river Po in Italy, but its provenance is still unexplained [12].

Dietary intake is considered as one of the main pathways for human exposure to PFASs, with food of animal origin being a primary source of exposure [4,13,14]. Bioaccumulation in aquatic and terrestrial food chains is one of the main processes that contribute to PFAS contamination in food [4]. Even though in recent years the literature on LC PFASs, especially PFOA and PFOS, in food and/or dietary intake has increased rapidly, data to estimate the intake via food of all the other PFASs remain scarce. This could be explained by the lack of reliable, sensitive analytical methods, which are required to detect PFASs in most food samples.

Currently, in Europe, there is no specific regulation on PFASs that establishes maximum residue levels in food but only a recommendation [15] that suggests limits of quantitation of 1 μg/kg and recovery rates in the 70–120% range for analytical methods. More recently, the network of the European Union Reference Laboratory (EURL) for Halogenated POPs in Feed and Food has published a useful guide, focusing on analytical parameters, in order to harmonize methods in the field of PFAS analysis in food and feed [16].

The European Food Safety Authority (EFSA) insists on the necessity of gathering as much data as possible to better understand consumers’ actual exposure to PFAS and to lower the detection limit of available analytical methods [4]. In fact, the tolerable weekly intake (TWI) has been recently drastically lowered to 4.4 ng/kg bw (body weight) as a cumulative value for PFOA, PFOS, perfluoro nonanoic acid (PFNA), and perfluoro hexane sulfonate (PFHxS) [4]. This urgently requires the development of a more sensitive and rapid analytical method for the simultaneous identification of both SC and LC PFASs in different food matrices.

Extraction and purification are crucial steps in sample preparation that need to be optimized during test development, accounting for the different chemical and physical properties of the analytes. Several studies have been published on detecting PFASs in different food matrices with optimized, often time-consuming, matrix-tailored methods. For example, Sadia et al. [17] recently published a method based on alkaline digestion, extraction, and clean-up with solid phase extraction and adsorption on granular carbon to detect only three PFASs (PFOS, PFOA, and PFHxS) in cow milk, butter, chicken meat, beef, and fish. Most of the published studies report different solvents for different matrices combined with weak anion exchange (WAX) solid phase extraction (SPE) as the most frequent clean-up procedure for detecting PFASs in different foods [18,19,20,21].

QuEChERS (quick, easy, cheap, effective, rugged, safe) is a well-established extraction/purification method currently used in wide-scope pesticide analysis in different food matrices [22]. The advantage of this technique relies on its speed and simplicity, involving micro-scale extraction with acetonitrile and extract purification using dispersive solid-phase extraction (d-SPE) [22]. Thanks to its versatility, QuEChERS has been recently adapted and optimized to detect different POPs in environmental and food samples [22], and thanks to its very quick and crude sample extraction and clean-up, it guarantees high throughput analysis for analytical laboratories.

The aim of this study was to develop and validate a fast, sensitive method for the simultaneous analysis of 14 PFASs (both SC and LC), including Gen X, C6O4, PFOA, PFOS, PFNA, and PFHXs, universally applicable to different foods of animal origin, i.e., meat, liver, egg, and milk. The QuEChERS extraction and clean-up method in combination with liquid chromatography coupled to mass spectrometry (LC-MSMS) was employed for this purpose and compared to the classical clean-up method based on WAX SPE previously used in our laboratory.

## 2. Results

### 2.1. Comparison between WAX SPE and QuEChERS Methods

To compare WAX SPE and QuEChERS, seven replicates of fish muscle, bovine muscle, and egg, each spiked to the level of 100 ng/kg with PFASs, were processed and analyzed. Figure 1, Figure 2 and Figure 3 show the comparisons in terms of recovery and repeatability between the two different approaches.

### 2.2. QuEChERS Method Validation

To ensure the adequate identification and quantification of the target compounds, the parameters of specificity, linearity, matrix effect, trueness, precision, LOD, LOQ, and measurement uncertainty were studied.

Specificity was assessed in blank matrices verifying the absence of a signal higher than 30% of the LOQ level (an example for bovine muscle is shown in Figure 4 and Figure 5 and in Appendix A). Taurodeoxycholic acid (TDCA), one of the most well-known PFOS-interfering substances found in matrices of animal origin (mainly eggs and offal) [16], was separated by chromatography.

Linearity was achieved for all the tested compounds; in fact, all the calibration curves met the acceptability criteria of a correlation coefficient (Pearson’s R) better than 0.98, and the deviation of back-calculated concentration from the true concentration of each calibration point was <20%.

No significant matrix effect was observed for any of the tested compounds in any of the different tested matrices (hen eggs, cow’s milk, bovine liver, fish muscle, swine muscle, and bovine muscle). In fact, by comparing, at the same concentrations, the slopes obtained in solvent calibration with the corresponding matrix-matched calibrations, all matrix results ranged between 80 and 120% of the solvent calibration results, regardless of the analyzed matrix.

Table 1, Table 2, Table 3, Table 4 and Table 5 summarize the recovery, precision, LOQ, LOD, measurement uncertainties, and matrix effect studied during the validation session for the different tested matrices. The validation parameters obtained were all compliant with the specific requirements reported in the EURL Guidance Document [16].

### 2.3. Analysis of Food Samples

The validated methods were applied to the analysis of 63 commercially available food samples. In 42 of the 63 samples, residual concentrations of PFASs were detected (Table 6). PFBA, which has only one specific MSMS transition, was also verified using a high-resolution MS method.

## 3. Discussion

Several studies have demonstrated that dietary intake is the main route of human exposure to PFASs, with matrices of animal origin having the main contribution [4]. The aim of this paper was to develop a fast and sensitive method easily applicable for routine analysis of PFASs in different matrices of animal origin.

Our previous in-house validated methods consisted of the WAX SPE extraction/clean-up approach [23], in line with other published studies [18,19,20,21]. The main drawback of this method was the length of analysis and, thus, the low daily throughput. It usually took two working days to process about 20 samples per analyst and to obtain a clean extract ready for analysis by LC-MSMS. For this reason, we decided to try a different approach and take advantage of the fast QuEChERS extraction/clean-up method [22] to shorten the analysis duration.

The first preliminary test highlighted comparable results in terms of recovery and repeatability for the two different procedures (Figure 1, Figure 2 and Figure 3). In fact, for all three tested matrices (fish muscle, bovine muscle, and eggs) the corresponding spiked samples at 100 ng/kg had a recovery in the range 80–120% for all compounds, except for the egg matrix where PFHxA and PFNA had the worst recoveries (67.7% and 62.7%, respectively) with the SPE procedure. As to repeatability, all the tested analytes/matrices had a CV% below 20% with both clean-up procedures.

Therefore, we decided to apply the QuEChERS procedure for validation purposes. Thanks to this different approach, about 30 samples per analyst are ready within one working day for LC-MSMS analysis. Good linear responses were obtained for all 14 compounds in the range of 125–7500 ng/L in the extracts (corresponding to 25–1500 ng/kg in the samples).

No significant matrix effects were observed for any of the tested compounds in all the different tested matrices (egg, milk, liver, and muscles) (Table 5). This allows the use of standard solution calibration curves instead of matrix-matched calibration ones. Thus, the concomitant analysis of different samples/matrices in the same analytical batch further reduced the time required for analysis in routine work (compared with our laboratory’s previous method).

As to trueness and precision, they met the acceptability criteria of 80–120% and ≤20%, respectively, and they comply with parameters reported in the EURL Guidance Document [16], regardless of the analyzed matrix, and for all the tested levels, as shown in Table 1 and Table 2.

LOQ and LOD revealed a sensitive method, by far more sensitive than the requirement of the Commission Recommendation 2010/161 [15] and in line with the Required LOQ reported in the EURL Guidance Document [16]. In fact, the LOD ranges obtained by the calibration approach were 7.78–16.35 ng/kg, 8.26–34.01 ng/kg, 6.70–33.65 ng/kg, and 5.92–19.07 ng/kg for milk, liver, egg, and muscle, respectively (Table 3). As to LOQ, it was fixed to 50 ng/kg for all the compounds except for GenX and C6O4 (for which it was equal to 100 ng/kg) for all the studied matrices. As to measurement uncertainty, it was <50% for all compound/matrix combinations.

To further assess the analytical performance of the method, our laboratory took part in a proficiency test organized by the European Union Reference Laboratory for Halogenated POPs in Feed and Food, consisting of the analysis of a real PFDA-, PFUnDA-, PFDoDA-, and PFOS-contaminated fish fillet (EURL-PT-POP_2001-FI). The performance of participating laboratories was assessed by calculating z-scores according to ISO 13528:2015 and the International Harmonized Protocol for the Proficiency Testing of Analytical Chemistry Laboratories [24]. The z-scores were satisfactory because they ranged between 0.7 and 1.2.

In general, the method complies with the recently proposed EURL requirements [16] and can be used to better study consumers’ exposure to PFASs through food. The methods were easily applicable to commercially available samples (Appendix A). In total 61 samples were analyzed, consisting of 34 livers, 6 eggs, 17 muscles, and 6 milk samples (Appendix A). Preliminary results revealed the absence of PFASs (< LOD) in all the analyzed milk samples. PFASs occurred (concentrations > LOD) in 42 of the 63 analyzed samples (Table 4), with livers (34 out of 34 samples) and eggs (6 out of 6 samples) being the most contaminated matrices.

LC compounds such as PFNA, PFDA, PFOS, and PFUnA were the most common contaminants. In particular, PFOS was the most abundant compound, present in all the liver samples. As to muscles, LC compounds were detected only in the fish samples (3 out of 17 muscle samples). These limited preliminary results were in line with previously published monitoring studies by EFSA [4].

The SC PFAS-like compound, PFBA, was found in all the egg samples. This result was in line with previously published studies conducted both on eggs from contaminated areas [25,26] and on commercial eggs [27]. SC PFASs, in fact, are less studied than the corresponding LC compounds [5].

These results highlight the necessity of implementing monitoring studies to determine PFAS concentrations and distributions in food.

## 4. Materials and Methods

### 4.1. Chemicals

In the present study 14 PFAS were analyzed, divided into SC PFASs-like and LC PFASs: perfluoro butanoic acid (PFBA), perfluoro pentanoic acid (PFPeA), perfluoro hexanoic acid (PFHxA), perfluoro heptanoic acid (PFHpA), perfluoro ottanoic acid (PFOA), perfluoro nonanoic acid (PFNA), perfluoro decanoic acid (PFDA), perfluoro undecanoic acid (PFUnA), perfluoro dodecanoic acid (PFDoA), perfluoro butane sulfonate (PFBS), perfluoro hexane sulfonate (PFHxS), perfluoro octano sulfonate (PFOS), 2,3,3,3-tetrafluoro-2-(heptafluoropropoxy)propanoic acid (GenX), and difluoro{[2,2,4,5-tetrafluoro-5-(trifluoromethoxy)-1,3-dioxolan-4-yl]oxy}acetic acid (C6O4).

The corresponding labeled standards were used for quantification purposes: perfluoro-n-[13C4] butanoic acid (MPFBA), perfluoro-n-[13C5] pentanoic acid (M5PFPeA), perfluoro-n-[1,2,3,4,6–13C5] hexanoic acid (M5PFHxA), perfluoro-n-[1,2,3,4–13C4] heptanoic acid (M4PFHpA), perfluoro-n-[13C8] octanoic acid (M8PFOA), perfluoro-n-[13C9] nonanoic acid (M9PFNA), perfluoro-n-[1,2,3,4,5,6–13C6] decanoic acid (M6PFDA), perfluoro-n-[1,2,3,4,5,6,7–13C7] undecanoic acid (M7PFUdA), perfluoro-n-[1,2–13C2] dodecanoic acid (M2PFDoA), perfluoro-1-[1,3,4–13C3] butansolfonate (M3PFBS), perfluoro-1-[1,3–13C3] esanosolfonate (M3PFHxS), and perfluoro-1-[13C8] octansolfonate (M8PFOS).

A solution containing a mixture of all the compounds (except GenX and C6O4), each at concentrations of 2 µg/mL, was purchased from Wellington Laboratories (Guelph, Canada). A mixture solution of the corresponding labeled standard, with each compound at a concentration of 2 µg/mL, was purchased from Wellington Laboratories (Guelph, ON, Canada).

GenX and C6O4 solutions, each at a concentration of 50 µg/mL, were purchased from Wellington Laboratories (Guelph, ON, Canada).

Acetonitrile (ACN) (UltraLC-MSgrade) and methanol (MeOH) (UltraLC-MSgrade) were purchased from VWR Chemicals (Radnor, PA, USA); ammonium acetate (approx. 98%) was purchased from Sigma-Aldrich (St Louis, MO, USA). Sodium acetate trihydrate (≥99.5%) and ammonium hydroxide (33% *v*/*v*) were supplied by Honeywell-Fluka (Charlotte, NC, USA). Solid phase extraction (SPE) was conducted with Oasis WAX cartridges (6 cc, 500 mg, 60 μm, Waters, Milford, MA, USA).

A Milli-Q-Plus ultrapure water system from Millipore (Bedford, MA, USA) was used to prepare Milli-Q water for preparation of samples and standards.

QuEChERS extraction salts (4 g Na_2_SO_4_, 1 g NaCl, 1 g trisodium citrate dehydrate, and 0.5 g disodium hydrogen citrate sesquihydrate) and dispersive SPE (150 mg di-primary secondary amine (PSA), 150 mg C18, and 900 mg of MgSO_4_) were supplied by Restek (Bellefonte, PA, USA).

Two intermediate mix solutions of all 14 PFASs at 10 and 100 ng/mL, respectively, were prepared by mixing the appropriate amounts of the corresponding solutions and diluting with HPLC-grade methanol.

An intermediate mix solution of the 12 labeled compounds, each at a concentration 10 ng/L, was prepared by appropriate dilution with HPLC-grade methanol of the pristine 2 µg/mL mix solution (Wellington Laboratories).

### 4.2. QuEChERS Sample Preparation

For egg, liver, muscle, and milk samples, 5 g of homogenized matrix was fortified with 50 µL of mass-labeled PFAS mixture at 10 µg/L, mixed with 10 mL of MQW and shaken on a vortex-mixer IKA Vibrax VXR (Staufen, Germany) for 1 min. After the addition of 10 mL of acetonitrile, samples were further shaken on an automatic stirrer (Genogrinder™, Spex^®^ Sample PREP, Stanmore, UK) at 25 Hz for 3 min.

To induce phase separation and pesticide partitioning in the organic phase, QuEChERS extraction salts were added. The tubes were then closed and shaken again on an automatic stirrer at 25 Hz for 1 min and centrifuged for 5 min at 6000× *g* (Eppendorf Centrifuge 5810, Amburg, Germany).

All the extracts were submitted to d-SPE clean-up. In detail, the extracts were transferred into a 15 mL plastic tube containing PSA, C18, and MgSO_4_, shaken on an automatic stirrer at 25 Hz for 3 min, and then centrifuged for 10 min at 6000× *g*.

Finally, 3.5 mL of extract was evaporated until dry under a gentle stream of N_2_ at 40 °C. The dried residue was dissolved in 0.35 mL of 10 mM ammonium acetate in 80% MeOH and 20% ACN. The final solution was transferred into vials for analysis by LC-MSMS.

### 4.3. SPE Sample Preparation

For egg and muscle matrices, 5 g of the homogenized matrix was fortified with 50 µL of mass-labeled PFAS mixture (10 µg/L). Subsequently, 20 mL of MeOH was added to each fortified matrix. Extraction was performed using an automatic agitator (Genogrinder™, Spex^®^ Sample PREP, Stanmore, UK) operating at 1500 rpm for 2 min. Extracts were then centrifuged for 10 min at 8500× *g* at 4 °C (Eppendorf Centrifuge 5810, Amburg, Germany). Each supernatant was then filtered through a filter paper (ashless/black ribbon 125 mm, particle filtration size of 12–25 μm; Whatman, Maidstone, UK), transferred to a new tube, and 20 mL of Milli-Q water was added. Cartridges for SPE were conditioned with 5 mL of MeOH and 4 mL of Milli-Q water. Afterward, each diluted extract was passed through a cartridge, which was then washed with 4 mL of 25 mM (pH 4) acetate buffer and dried. PFASs were eluted from the cartridge with MeOH containing 5% ammonium hydroxide. The collected extract was evaporated to dryness under a gentle stream of N_2_ at 40 °C. The dried residue was dissolved in 1 mL of 10 mM of ammonium acetate in 80% MeOH and 20% ACN. The final solution was transferred into a vial for analysis by LC-MSMS.

### 4.4. Instrumental Analysis 

LC-MSMS analysis was performed on a Shimadzu LC system (Kyoto, Japan) coupled to an API 6500 AB SCIEX (Framingham, Massachusetts, USA) Triple Quadrupole (QQQ), operating with electrospray ionization (ESI) in negative mode. An XTerra^®^ MSC18 (2.1 × 100 mm) 5 µm column was used as a delay column prior to the injector in order to isolate and delay potential PFAS traces coming from the LC system. A 5 µL aliquot of the sample was injected into a Waters Aquity UPLC BEH Shield RP 18 (2.1 × 100 mm), 1.7 µm column. The mobile phases were water with 10 Mm ammonium acetate (FM A) and MeOH/ACN (80/20) with 10 mM ammonium acetate (FM B). The gradient applied was: 0–0.1 min of 5% FM B, then FM B was increased linearly to reach 95% FM B at 8 min and kept for two minutes. Finally, the concentration of FM B was decreased to 5% at between 10 and 10.5 min, and the column was equilibrated to initial conditions until 14 min. The flow rate was 0.3 mL/minute, and the column oven temperature was set at 30 °C.

The gas temperature and the ion spray voltage of the QQQ were kept at 350 °C and −2500 V, respectively.

Ions were monitored using a multiple reaction monitoring (MRM) mode. Transitions for each target analyte are reported in Appendix A.

The isotope dilution method was applied for the analysis of the samples. In particular, the corresponding labeled compounds for each of the 12 PFAS were used as internal standards to calculate the relative response factor of the corresponding native compound and to confirm the retention time (RT). For GenX and C6O4, M5PFHxA was used as an internal standard.

Analyst software AB SCIEX (Framingham, MA, USA) was used to control the LC-MS system, and Multiquant 3.0.2 software (Framingham, MA, USA) was used to quantify analytes.

### 4.5. Method Validation

To date, the European Commission has not set maximum limits for PFASs in food, but quite recently, the network of the European Union Reference Laboratory (EURL) for Halogenated POPs in Feed and Food published a EURL Guide with specific rules to assess method performances for their determination in food [16]. This EURL Guide was strictly followed in our study. Moreover, Regulation 333/2007 [28] laying down the methods of sampling and analysis for the official control of the levels of some contaminants in foodstuffs and the guidelines laid down by Document N° SANTE/11312/2021 [29], stipulating the conventional validation approach required for quantitative confirmation, were also followed to evaluate the fitness for purpose of the analytical method developed here.

The following parameters were evaluated: specificity, linearity, matrix effect, trueness, precision, LOQ, LOD, and measurement uncertainty. 

Blank matrices of egg, muscle (bovine, swine, and trout), liver (bovine), and cow’s milk used for validation purposes were recovered from the laboratory repository and tested beforehand to verify the absence of contamination.

Specificity was assessed in each blank matrix by verifying the absence of a signal higher than 30% of the LOQ level.

Linearity was studied by means of calibration curves constructed with seven levels (including zero level) within the range of 125–7500 ng/L (corresponding to 25–1500 ng/kg in the sample). Calibration curves were built by plotting the instrument signal versus the analyte concentration. Linear regression analysis was carried out, and the linear calibration model was verified by correlation coefficients (Pearson’s R) better than 0.992 and by the Mandel test. Deviation from back-calculated concentration from true concentration was less or equal to 20%. 

The matrix effect was studied by comparing the slope obtained in solvent calibrations with the corresponding matrix-matched calibration at the same concentrations, verifying the signal suppression or enhancement (not more than 20%).

Trueness and precision were estimated by analyzing seven replicates at four concentration levels for all the target compounds (50–100–500–1000 ng/kg) except for GenX and C6O4, for which three levels were tested (100–500–1000 ng/kg).

LOD was determined according to the JRC Technical Report “Guidance Document on the Estimation of LOD and LOQ for Measurements in the Field of Contaminants in Feed and Food” [30] via the calibration approach, as defined by the following equation:(1)$$x_{LOD}=(t_{\alpha ,\upsilon }+t_{ß,\upsilon })\cdot \frac{s_{y,x}}{b}\cdot \sqrt{\frac{1}{m}+\frac{1}{p\cdot q}+\frac{{\bar{x^{\;}}}^{2}}{\sum_{i=1}^{n}{\left(x_{i-}\bar{x}\right)}^{2}}}$$
*p* number of calibration levels;*q* number of replicate analyses per calibration level;t_α_, value from t-distribution for probability level α = 0.05 (one-sided test) andυ = (p × q) −2 degrees of freedom;t_ß,υ_ value from t-distribution for probability level ß = 0.05 (one-sided test) andυ = (p × q) −2 degrees of freedom;s_y,x_ standard deviation of the residuals;*b* slope of the calibration curve;*m* number of replicate analyses of the test sample;$\bar{x}$ content value corresponding to the mean calibration level;*x_i_* content value of the analyte at calibration level i.

LOQ was identified as the lowest analyte content level studied in the validation process at which precision (CV%_repeatability_ ≤ 20%) and trueness (recovery 80–120%) were within the limits established by guidance documents [16,29].

Measurement uncertainty was determined according to NMKL PROCEDURE No. 5 (2019) “Estimation and expression of measurement uncertainty in chemical analysis” [31], as determined by the following equation:$$u_{c}=\sqrt{{\left(R_{w}\right)}^{2}+u{\left(bias\right)}^{2}}$$
*R_w_* within laboratory reproducibility*u(bias) u (bias)*$=\sqrt{RMS_{bias}^{2}+u{\left(C_{recovery}\right)}^{2}}$*RMS_bias_*$RMS_{bias}=\sqrt{\frac{\sum {\left(bias_{i}\right)}^{2}}{n}}$*u(C_recovery_)* $u\left(C_{recovery}\right)=\sqrt{{\left(\frac{U_{std}}{2}\right)}^{2}+\left({\left(\frac{Bias_{pipette}}{\sqrt{3}}\right)}^{2}+u_{r,Pipette}^{2}\right)}$*U_std_* 95% confidence interval for the concentration of the standard;*Bias_pipette_* Volume specification for maximal bias for the pipette;*u_r,pipette_* Volume specification for maximal repeatability for the pipette.

Identification was achieved by the ion ratio of the secondary mass transition response relative to the primary mass transition response. Molecular ions (as deprotonated molecules) were selected as parent ions. The retention times were also recorded for each compound for identification purposes.

### 4.6. Analysis of Real Samples and Quality Controls

The method was applied to the analysis of 63 food samples (at least 5 samples per matrix type): 34 livers, 6 eggs, 17 muscles, and 6 milk samples. 

All the samples were obtained from the laboratory repository. Detailed information about these food samples is reported in Appendix A.

Two negative QC samples (one with all processing reagents but without a matrix sample, and a second consisting of fully processed blank matrix samples; eggs, muscle, and liver), and two positive QC samples (two blank matrix-matched samples spiked at 50 ng/kg per matrix) were processed and analyzed in each analytical batch to verify method performance and the absence of undesired contamination.

## Figures and Tables

**Figure 1 molecules-27-07899-f001:**
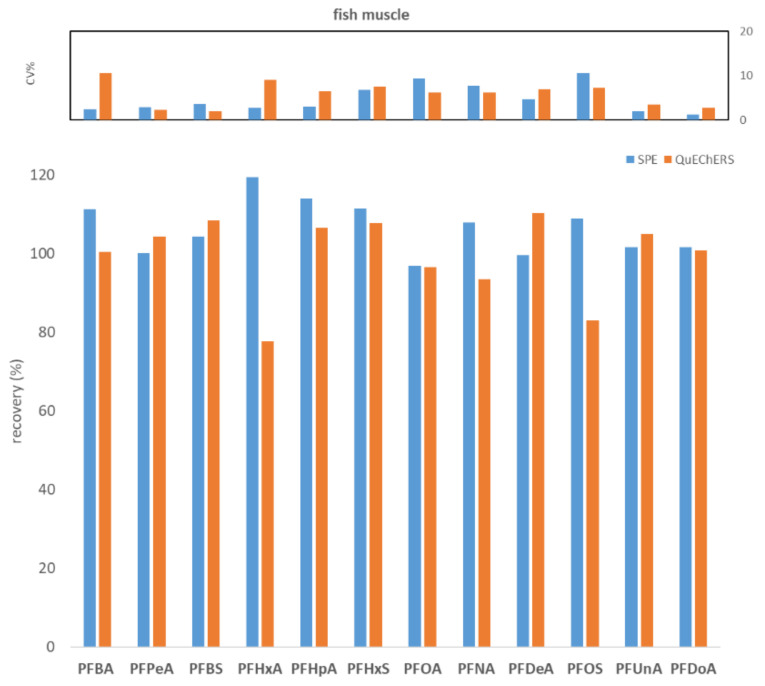
Comparison between SPE and QuEChERS in terms of repeatability (calculated as CV% of seven replicates at 100 ng/kg) and recovery (calculated as the average of seven replicates at 100 ng/kg) for the fish muscle matrix.

**Figure 2 molecules-27-07899-f002:**
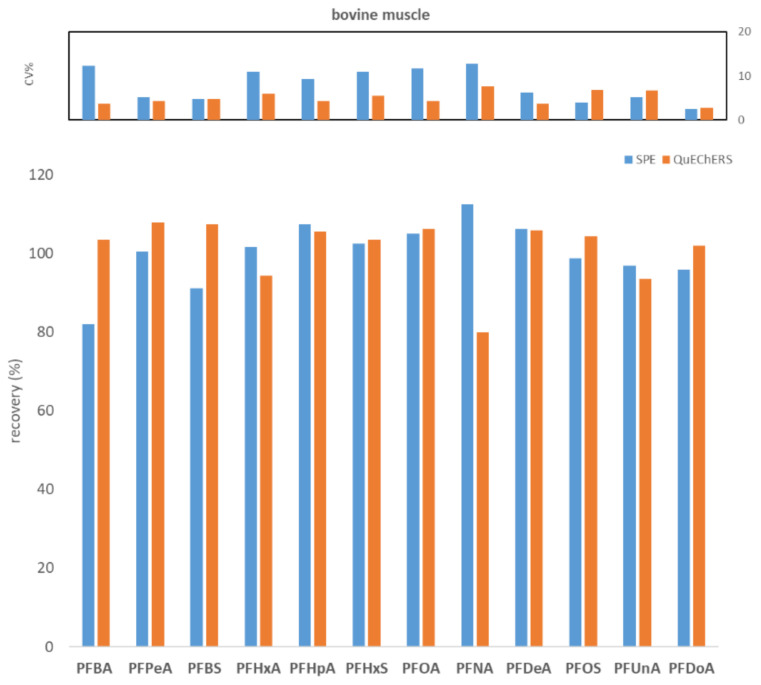
Comparison between SPE and QuEChERS in terms of repeatability (calculated as CV% of seven replicates at 100 ng/kg) and recovery (calculated as the average of seven replicates at 100 ng/kg) for the bovine muscle matrix.

**Figure 3 molecules-27-07899-f003:**
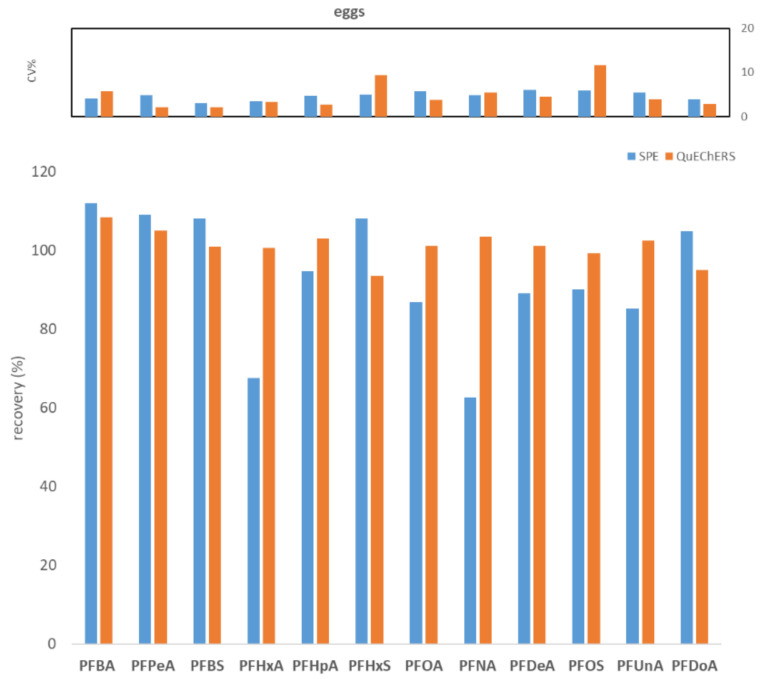
Comparison between SPE and QuEChERS in terms of repeatability (calculated as CV% of seven replicates at 100 ng/kg) and recovery (calculated as the average of seven replicates at 100 ng/kg) for the hen eggs matrix.

**Figure 4 molecules-27-07899-f004:**
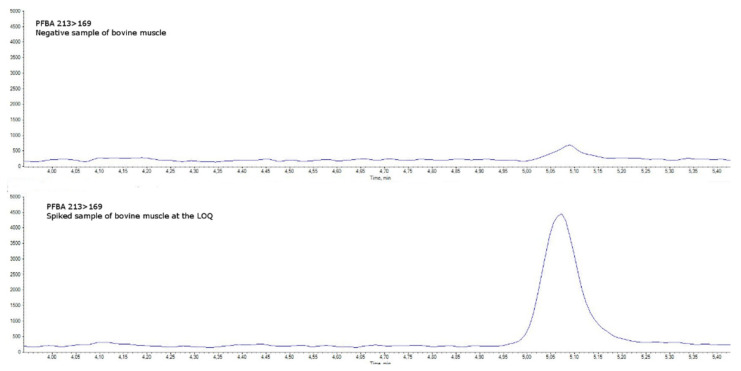
(PFBA quantitative transition 213 > 169). Comparison between a negative sample of bovine muscle (upper chromatogram) and a spiked sample of bovine muscle at the limit of quantification (lower chromatogram).

**Figure 5 molecules-27-07899-f005:**
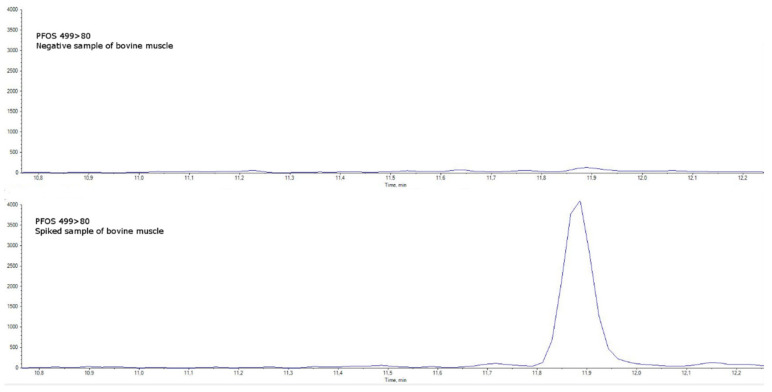
(PFOS quantitative transition 499 > 80). Comparison between a negative sample of bovine muscle (upper chromatogram) and a spiked sample of bovine muscle at the limit of quantification (lower chromatogram).

**Table 1 molecules-27-07899-t001:** Method recovery and repeatability in hen eggs, cow’s milk, and bovine liver intraday measurements (one blank sample spiked with four different concentrations, seven replicates for each concentration).

Compound	Nominal Concentration (ng/kg)	Recovery (%) (*n* = 7)			Precision (CV %) (*n* = 7)		
		Hen Eggs	Cow’s Milk	Bovine Liver	Hen Eggs	Cow’s Milk	Bovine Liver
PFBA	50 *	96.9	97.3	103.0	7.6	3.9	3.8
	100	108.5	106.5	98.6	5.7	1.6	8.3
	500	106.5	102.6	101.5	2.6	2.5	2.4
	1000	104.6	103.2	103.3	0.9	1.7	3.8
PFPeA	50 *	105.0	99.8	99.2	1.2	4.0	5.4
	100	105.1	102.8	101.0	2.1	3.4	6.0
	500	104.0	103.8	101.8	2.2	1.6	2.5
	1000	103.1	105.8	97.5	0.7	1.6	4.4
PFBS	50 *	99.2	103.0	102.3	1.2	6.1	4.8
	100	101.0	107.5	105.1	2.1	5.0	4.6
	500	101.8	103.3	108.2	2.2	3.7	5.0
	1000	97.5	104.8	105.0	0.7	2.6	3.9
PFHxA	50 *	95.5	115.8	106.2	1.8	6.7	7.6
	100	100.7	112.4	105.8	3.4	4.5	3.9
	500	104.7	105.6	107.7	2.3	6.0	4.3
	1000	103.6	105.7	105.1	1.5	1.2	3.0
PFHpA	50 *	106.0	97.4	106.0	2.2	5.1	9.8
	100	103.0	104.3	103.0	2.7	4.7	5.7
	500	103.0	110.2	103.0	2.8	1.9	3.3
	1000	107.8	110.1	107.8	4.1	1.4	8.7
PFHxS	50 *	91.4	100.5	92.7	8.4	5.5	7.9
	100	93.5	110.7	104.7	9.4	5.0	7.22
	500	99.5	104.6	96.9	9.9	5.1	5.6
	1000	101.9	104.4	97.9	11.5	6.4	13.7
PFOA	50 *	97.5	101.8	105.5	3.8	3.9	3.6
	100	101.1	102.5	103.4	3.8	4.2	6.8
	500	106.0	101.6	107.2	1.0	1.8	9.4
	1000	107.7	105.8	102.4	2.5	3.1	4.8
PFNA	50 *	111.7	121.2	111.7	6.5	8.9	6.4
	100	103.6	121.8	103.6	5.4	3.1	4.5
	500	106.7	106.3	106.7	1.6	2.2	3.6
	1000	105.1	108.3	105.1	2.7	3.1	8.3
PFDeA	50 *	91.8	111.0	115.1	5.3	2.3	10.1
	100	101.2	112.0	110.9	4.6	1.6	2.6
	500	110.1	102.1	108.5	3.8	1.4	3.3
	1000	121.1	106.6	104.1	5.3	3.1	6.8
PFOS	50 *	86.7	93.9	103.1	5.6	3.8	7.7
	100	99.3	107.0	93.2	11.6	9.0	15.4
	500	107.8	100.3	94.1	6.1	6.3	8.4
	1000	105.0	105.5	81.0	3.7	5.0	11.7
PFUnA	50 *	105.8	109.2	105.8	2.8	3.9	4.4
	100	102.5	104.2	102.5	4.0	2.9	2.0
	500	104.8	101.2	104.8	2.3	2.5	3.2
	1000	107.1	104.8	107.1	3.8	2.5	4.7
PFDoA	50 *	87.8	105.0	104.6	1.8	2.5	3.0
	100	95.0	107.3	103.8	2.9	3.0	3.8
	500	100.3	103.0	106.3	1.8	1.8	2.0
	1000	103.2	103.9	105.8	1.6	2.4	3.9
GenX	100 *	94.9	103.8	94.9	13.0	10.9	14.1
	500	92.4	91.3	92.4	10.0	5.3	8.1
	1000	93.7	94.6	93.7	9.0	4.2	16.1
C6O4	100 *	107.9	109.2	94.4	10.1	3.7	7.3
	500	94.9	97.3	84.6	14.0	10.5	19.3
	1000	111.1	97.4	94.0	9.5	8.8	16.5

* limit of quantification.

**Table 2 molecules-27-07899-t002:** Method recovery and repeatability in fish muscle, swine muscle, and bovine muscle samples intraday measurements (one blank sample spiked with four different concentrations, seven replicates for each concentration).

Compound	Nominal Concentration (ng/kg)	Recovery (%) (*n* = 7)			Precision (CV %) (*n* = 7)		
		Fish Muscle	Swine Muscle	Bovine Muscle	Fish Muscle	Swine Muscle	Bovine Muscle
PFBA	50 *	91.9	83.6	100.0	13.0	12.2	4.9
	100	100.4	89.4	103.6	10.5	11.8	3.7
	500	92.8	96.8	111.5	3.5	2.0	1.5
	1000	102.2	101.2	111.0	5.7	3.8	1.2
PFPeA	50 *	105.3	111.6	119.0	4.5	2.8	2.8
	100	104.3	103.3	107.9	2.3	4.3	4.3
	500	97.2	101.2	109.4	3.6	1.3	1.5
	1000	103.1	105.6	108.7	5.1	4.5	2.38
PFBS	50 *	109.2	107.3	100.1	4.6	2.8	6.6
	100	108.3	98.7	107.4	2.0	5.8	4.7
	500	99.3	98.6	111.3	5.2	2.9	3.6
	1000	103.7	98.5	108.4	5.5	4.9	2.1
PFHxA	50 *	80.0	90.4	77.8	9.6	5.5	14.6
	100	77.7	90.2	94.4	9.11	5.8	5.9
	500	93.9	102.0	105.6	3.2	2.8	2.4
	1000	106.6	102.3	108.8	6.0	9.8	4.2
PFHpA	50 *	110.3	105.5	93.3	6.9	4.3	15.4
	100	106.5	98.4	105.5	6.5	7.7	4.3
	500	95.1	100.6	110.2	7.0	3.9	2.1
	1000	100.9	105.4	115.2	8.7	7.2	5.2
PFHxS	50 *	107.1	107.2	106.3	4.8	5.4	6.8
	100	107.7	103.4	103.6	7.6	3.7	5.5
	500	105.9	100.8	108.6	2.1	3.9	6.7
	1000	105.6	96.6	114.2	7.0	12.9	8.3
PFOA	50 *	89.6	96.4	113.7	6.3	3.0	6.1
	100	96.5	98.2	106.3	6.1	2.1	4.3
	500	94.9	100.9	115.4	6.9	2.5	3.0
	1000	105.9	104.4	113.3	6.3	5.9	2.5
PFNA	50 *	67.9	103.2	82.4	8.7	5.2	11.1
	100	93.4	104.2	80.0	6.1	7.3	7.6
	500	100.9	106.1	104.8	3.6	3.1	4.8
	1000	106.4	110.8	110.9	6.5	6.0	4.7
PFDeA	50 *	120.2	90.3	111.9	4.8	2.8	7.0
	100	110.3	98.9	105.9	6.9	2.8	3.7
	500	95.2	102.7	116.6	6.4	1.1	2.0
	1000	102.2	110.9	120.2	9.4	3.7	3.4
PFOS	50 *	102.9	111.8	102.1	14.6	10.2	16.2
	100	83.0	104.3	104.4	7.2	8.3	6.8
	500	86.3	103.9	105.1	3.7	7.6	5.9
	1000	103.6	105.9	110.4	12.7	7.8	9.6
PFUnA	50 *	107.9	112.5	82.3	4.4	2.2	4.5
	100	105.0	107.1	93.5	3.4	5.0	6.7
	500	98.1	100.9	109.8	4.7	2.5	1.8
	1000	105.7	105.9	113.5	6.0	4.5	1.5
PFDoA	50 *	103.7	106.6	105.3	2.2	3.5	1.4
	100	100.7	104.4	102.1	2.7	3.5	2.8
	500	91.1	99.0	111.1	4.2	3.5	2.9
	1000	94.7	102.9	110.5	5.4	5.6	2.2
GenX	100 *	79.9	80.0	57.2	11.2	15.9	5.9
	500	88.3	69.2	66.8	4.8	4.8	6.6
	1000	97.5	74.4	81.1	6.2	8.6	2.2
C6O4	100 *	103.8	82.1	73.5	10.6	13.1	7.1
	500	99.1	72.9	85.1	5.6	4.5	15.6
	1000	96.8	79.8	85.8	9.3	13.9	12.2

* limit of quantification.

**Table 3 molecules-27-07899-t003:** Limit of detection (LOD) calculated according to the calibration approach.

		Matrix		
Analyte	Milk	Liver	Egg	Muscle
	LOD ng/kg	LOD ng/kg	LOD ng/kg	LOD ng/kg
PFBA	11.01	8.43	7.20	8.05
PFPeA	10.47	8.72	8.65	5.78
PFBS	12.77	10.78	14.26	6.87
PFHxA	8.38	8.65	7.50	14.76
PFHpA	4.43	9.09	8.28	5.81
PFHxS	4.68	7.84	10.10	10.39
PFOA	4.86	6.31	8.03	9.01
PFNA	4.94	9.27	19.17	8.07
PFDeA	7.79	6.40	18.45	9.19
PFOS	4.75	6.27	9.08	17.33
PFUnA	10.15	4.66	11.35	11.49
PFDoA	6.95	7.33	6.70	3.69
GenX	7.14	13.11	13.75	12.98
C6O4	9.30	19.84	8.21	8.70

**Table 4 molecules-27-07899-t004:** Expanded relative measurement uncertainties calculated according to NMKL procedure No. 5 (2019).

Analyte	Matrix			
	Milk	Liver	Egg	Muscle
	(%)	(%)	(%)	(%)
PFBA	14.66	29.28	39.42	25.22
PFPeA	16.56	45.40	29.60	29.52
PFBS	23.36	41.44	24.18	18.48
PFHxA	28.80	40.80	38.10	46.38
PFHpA	16.86	32.94	24.44	33.94
PFHxS	19.26	32.70	23.38	19.34
PFOA	14.34	26.16	23.66	33.24
PFNA	31.02	27.12	45.92	47.60
PFDeA	23.56	49.44	17.28	31.56
PFOS	20.66	46.84	30.94	38.22
PFUnA	23.04	19.68	43.24	32.96
PFDoA	19.64	31.06	25.12	29.58
GenX	32.56	32.30	47.32	45.52
C6O4	23.44	34.06	45.86	37.02

**Table 5 molecules-27-07899-t005:** Matrix effect expressed as the percentage difference of response (slope of the calibration curves) from the standard in the matrix extract and standard in solvent.

Compound	Milk	Liver	Egg	Muscle
PFBA	−6%	−13%	−2%	−3%
PFPeA	−11%	−15%	−11%	−9%
PFBS	−2%	−14%	−4%	−5%
PFHxA	−8%	−10%	−9%	−8%
PFHpA	−5%	−12%	−8%	−11%
PFHxS	−8%	−9%	−9%	−10%
PFOA	−7%	−8%	−6%	−7%
PFNA	−5%	−10%	−6%	−8%
PFDeA	−4%	−13%	−2%	−5%
PFOS	−5%	−5%	−4%	−6%
PFUnA	−8%	−13%	−7%	−10%
PFDoA	−5%	−8%	−6%	−8%
GenX	−9%	−11%	−8%	−10%
C6O4	−11%	−13%	−9%	−7%

**Table 6 molecules-27-07899-t006:** PFASs detected in real samples.

Matrix	Species/ Category	Sample ID	PFBA ng/kg (*n* = 2)	PFHXS ng/kg (*n* = 2)	PFOA ng/kg (*n* = 2)	PFNA ng/kg (*n* = 2)	PFDA ng/kg (*n* = 2)	PFOS ng/kg (*n* = 2)	PFUNA ng/kg (*n* = 2)	PFDOA ng/kg (*n* = 2)
Liver	Trout	L1	138	179		91		4215	126	28 *
	Trout	L2		192		117		4083	128	27 *
	Calf	L4					60	159		
	Calf	L5					41 *	100		
	Calf	L6				35 *	50	90	25 *	23 *
	Calf	L7				36 *	48 *	132	11 *	36 *
	Calf	L8		30 *		36 *	35 *	699		
	Calf	L9				68	64	301	30	
	Calf	L10				35 *	55	160	18	
	Calf	L11				36 *	56	149	19	
	Calf	L12				18 *	54	156	17	
	Calf	L13				19 *	29 *	53	42	
	Calf	L14				41 *	72	140	14	
	Bullock	L15				43 *	59	246	14	
	Bullock	L16				36 *	85	329	15	
	Bullock	L17				38 *	62	186		
	Bullock	L18					77	201		
	Bullock	L19				91	193	225	41 *	
	Bullock	L20				78	98	220	23 *	
	Bullock	L21				35 *	60	105	18 *	
	Bullock	L22				38 *	63	177	13 *	
	Bullock	L23				38 *	54	232	16 *	
	Bullock	L24				23 *				
	Bovine Adult	L25	123			284	409	1099	124	44 *
	Bovine Adult	L26	52			304	577	1362	203	71
	Bovine Adult	L27	75			262	446	1220	144	53
	Bovine Adult	L28	50			255	411	1195	128	54
	Bovine Adult	L29	84			132	388	1104	108	44 *
	Bovine Adult	L30	85			167	622	1609	179	58
	Bovine Adult	L31	83			137	314	1117	112	47 *
	Bovine Adult	L32	93			158	365	1092	119	47 *
	Swine	L33			39 *	73	55	2874	106	
	Swine	L34	102			28 *		63		
Eggs	Hens	E1	119							
	Hens	E2	25 *	40 *	25 *	60	150	910	71	170
	Hens	E3	46 *							
	Hens	E4	101							
	Hens	E5	110							
	Hens	E6	119							
Muscle	Trout	M4			15 *	34 *	70	360	89	94
	Trout	M5			10 *	36 *	80	331	86	93
	*Pangasius*	M6				16 *	34 *	99	69	

* Value above the limit of detection, but below the limit of quantification.

## Data Availability

Not applicable.

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
