# Peer review of "Fast and Sensitive Analysis of Short- and Long-Chain Perfluoroalkyl Substances in Foods of Animal Origin"

_molecules, 2022, doi:10.3390/molecules27227899_

Round 1
Reviewer 1 Report
Molecules-1945374.
Fast and sensitive analysis of short and long chain perfluoroal-2 kyl substances in foods of animal origin.
The authors propose a manuscript describing the development and validation of a global method of analysis of PFASs by U-HPLC -MS/MS. The introduction is well written and shows the state of the art and positions the subject of the manuscript well.
The major problem with this paper is with the experimental part.
Two U-HPLC columns were coupled (Page 15, Lines 349-350). One BEH Shield RP18 UPLC column and one BEH C18 UPLC column. The latter is presented as a guard column (line 350). It is surprising to use a guard column different from the main column, but especially here the guard column is a column in itself (2.1x100 mm). Moreover I did not know the columns 2.1x500 mm from Waters? in any case, it is impossible to couple 500 mm + 100 mm of stationary phase on a U-HPLC system with a flow rate of 0.3mL/min.
On this question, the authors must provide us a precise answer, check their column references if the dimensions are indeed that indicated and justify the use of 2 different stationary phases, for example by providing bibliographic references.
This manuscript, which presents an LC-MS/MS method, gives no chromatogram in result. At least one figure showing all the molecules separated in MRM mode for a median concentration should be provided to illustrate the development.
This could be done by simplifying FIGS. 1 to 6 by giving the repeatability CVs directly on the recovery histograms. There would thus be only 3 figures, leaving the place to one presenting a chromatogram.
Specific points:
Replace HPLC-MSMS or LC-MSMS with HPLC-MS/MS or LC-MS/MS throughout text
Figures 1-6 Page 3-6. The authors do not discuss the results concerning extraction yields and repeatability for the 2 methods evaluated and in the light of the literature. Why then provide these figures? It is necessary to discuss the results in relation to the other results of the literature and give the benefits of the QuEChERS extraction if any.
Linearity. Lines 145-148. In method validation, it is important to also compare the slopes of the lines and to discuss the linearity in relation to this slope and the range of concentrations selected. Compare these results to the literature.
Matrix Effect Line 149-154. A table with the matrix effect calculated for each molecule should be provided to identify whether there is a positive or negative effect on ionization.
Na2SO4 Line 300: Format formula
Table 1. Line 365: Retention time (minutes), delete the s for minute s
Reviewer 2 Report
In this paper, a sensitive and fast method has been developed and validated for the simultaneous determination of short and long chain PFAS in several food of animal origin. A QuEChERS extraction and clean-up of samples method along with LC-MSMS has been used as a novelty to obtain high throughput in the analysis. The obtained results by this clean up method were compared in term of recovery and repeatability with those obtained by a classical method based in SPE. The method has been exhaustively validated obtained results very favorable in most studied samples.This method could be universally applicable for high throughput routine analysis in analytical laboratories.
It would be very interesting to include optimization studies of the parameters involved in the extraction and cleaning method used, as well as to provide discussion in this regard, since this is a novelty in this method.
The number of figures should be reduced. Include only the most significant, for example those where the differences between the results of the studies compared were greater, the others can be commented in the text.
Tables should be reduced by including ranges of results.
Information summarized over Table 1 should be included in "Instrumental analysis" section. This table can be included in "Supplementary Material".
More information on the analysis of real samples should be included in the paper.
